# The Impact of Burnout Syndrome on Job Satisfaction among Emergency Department Nurses of Emergency Clinical County Hospital “Sfântul Apostol Andrei” of Galati, Romania

**DOI:** 10.3390/medicina58111516

**Published:** 2022-10-25

**Authors:** Cosmina-Alina Moscu, Virginia Marina, Liliana Dragomir, Aurelian-Dumitrache Anghele, Mihaela Anghele

**Affiliations:** 1Emergency Department of Hospital of Galati, 800201 Galati, Romania; 2Medical Department of Occupational Health, Faculty of Medicine and Pharmacy, “Dunărea de Jos” University of Galati, 800201 Galati, Romania; 3General Surgery Department, Faculty of Medicine and Pharmacy, “Dunărea de Jos” University of Galati, 800201 Galati, Romania; 4Clinical-Medical Department, Faculty of Medicine and Pharmacy, “Dunărea de Jos” University of Galati, 800201 Galati, Romania

**Keywords:** burnout syndrome, professional satisfaction, nurse, Emergency Department

## Abstract

*Background and Objectives*: Burnout syndrome is caused by a number of factors, including personal, organizational, and professional problems. Data from the literature reported a strong relationship between burnout syndrome and job satisfaction among emergency medical personnel. *Materials and Methods:* We studied a sample of 80 nurses working in the Emergency Department of Emergency Clinical County Hospital “Sfântul Apostol Andrei” of Galati Romania. Participants signed an informed consent and then completed a socio-demographic questionnaire and the MBI-HSS (Maslach Burnout Inventory–Human Services Survey) questionnaire to assess the level of burnout and JSS (Job Satisfaction Survey) to assess the level of professional satisfaction. The aim of this study was to measure the levels of burnout and satisfaction among nurses in the Emergency Department as well as the relationship between these two variables and a group of selected socio-demographic characteristics. *Results:* This study found that 36.25% of nurses reported a high level of burnout. Additionally, emotional exhaustion is directly proportional to professional experience and age. Participants also reported a sense of ambivalence and satisfaction with the workplace, but were satisfied with the nature of their work. *Conclusions:* The results of this study suggest the relationship between variables and this could be used to implement psychoactive intervention strategies at both individual and organizational levels, which could lead to a decrease in burnout levels. Burnout is a predictor of job satisfaction for Emergency Department nurses. Personal achievement was relatively commensurate with the nature of the work. Additionally, the increase in burnout among nurses is directly proportional to the nature of the work.

## 1. Introduction

Burnout or professional burnout is a state of physical and emotional exhaustion, in which recovery is difficult and occurs especially in people whose profession involves increased responsibility and emotional involvement [1].

Professional exhaustion syndrome is a manifestation of chronic stress characterized by emotional, mental and physical exhaustion, caused by occupational stress factors, and is represented by the three dimensions: Emotional exhaustion, depersonalization and low personal accomplishment. According to Maslach, the first stage of Burnout syndrome is emotional exhaustion, characterized by loss of motivation, emotional overload and perception of activity at work as a chore. Depersonalization, the second stage of Burnout syndrome is characterized by the appearance of impersonal feelings towards work or other people. The third stage is represented by personal accomplishment that involves the decrease in satisfaction and professional efficiency associated with a low self-esteem [2,3].

Burnout among nurses working in the Emergency Department is a serious worldwide problem and is associated with adverse workplace factors (intention to leave their job, stress at work, and verbal and physical aggression at work) [4,5].

According to the Romanian Ministry of Health order no.1706/2007 on the management and organization of Emergency Departments, medical staff in the Emergency Department are trained to intervene in pre-hospital emergencies, disaster medicine, patient triage and resuscitation [6].

The Emergency Department and prehospital medical services are extremely stressful environments, with medical staff being overworked both physically and mentally due to high workload, high patient flow with complex pathology, night shifts, high noise levels, bureaucracy, and prehospital work [7].

According to the specialized literature, nurses with high levels of burnout feel emotionally exhausted at the end of the day due to an imbalance between workload and occupational resources, with an inability to attend to the social and emotional needs of family members or patients [8].

Job satisfaction is a positive emotional state that workers experience in relation to their work [9]. Factors influencing job satisfaction are financial rewards, working conditions, professional capability and psychosocial relationships at work.

According to the literature, job satisfaction in health care is related to many factors: Optimal working conditions, the opportunity to actively participate in decision making and collective problem solving, as well as effective communication between staff and supervisors [10,11].

The aim of this study was to identify the level of Burnout syndrome among Emergency Department nurses, as well as the impact of Burnout syndrome on job satisfaction. Our hypothesis is that nurses have a high level of burnout, as well as a high level of job satisfaction. As secondary objectives, we identified the socio-demographic characteristics of the examined group and determined the correlation between the dimensions of the burnout syndrome and the collected socio-demographic data.

## 2. Material and Methods

### 2.1. Study Population and Study Protocol

We issued a total of 112 questionnaires, receiving 80 complete and valid questionnaires with a response rate of 71.42%.

The survey was anonymous, containing instructions on how to complete the socio-demographic data and questionnaire, as well as written informed consent to participate in the study.

This was a descriptive, cross-sectional study of nurses in the Emergency Department of the Emergency Clinical County Hospital “Sf. Ap. Andrei” Galati, Romania.

Data collection took place between March and April 2022. The professional categories included were nurses working in both the Emergency Department and in the prehospital. Exclusion criteria: Experience less than one-year, incomplete questionnaires and a lack of consent to participate in the study. The study was conducted in accordance with the World Medical Association Declaration of Helsinki, using a protocol approved by the administration of “Sfântul Apostol Andrei” County Emergency Clinical Hospital Galati, Romania. (5257/02.03.2021).

### 2.2. Instruments

To assess the level of Burnout Syndrome, we used the Maslach Burnout Inventory Human Services Survey (MBI-HSS), designed by Christina Maslach and Susan E. Jacksonin 1981, for people working in a wide range of occupations, including nurses, and containing 22 items and structured on three scales: Emotional exhaustion (9 items), depersonalization (5 items) and low personal accomplishment (8 items). Each item is given a Likert-type response in terms of the frequency with which the participant experiences these “traumas” [12] (Maslach et al., 2022). The score of each dimension can be calculated using two methods. The first method sums up the scores by obtaining scores for each dimension and the second method calculates the average of the scores for each dimension. This study adopted the first method of calculation, so for each scale, the sum of the points is calculated and a score is obtained. Thus, emotional exhaustion is high for a score of at least 27, depersonalization is high for a score of at least 10, while personal accomplishment is low for a score of 33 or less. According to other studies that measured the burnout syndrome of medical professionals, a high level of burnout syndrome was defined with a high score on the dimension of emotional exhaustion (≥27) and/or depersonalization (≥10) [13,14,15,16]. This instrument has also been applied to a population of Romanian origin [17]. For this study, we determined the Cronbach’s alpha coefficient with values 0.79 for MBI-HSS.

This instrument was used to measure the professional exhaustion of the medical staff in Romania, and the translation was performed by Horia Pitariu and Maria Cseh on 15 June 1996, with the consent of the Consulting Psychologists Inc. publishing [18,19,20].

To assess the level of job satisfaction, we used the Job Satisfaction Survey (JSS) questionnaire designed by Paul E. Spector in 1994, which contains 36 items, structured into 9 scales to assess employee attitudes toward the job and aspects of the job. The nine aspects of the JSS are: Compensation, promotion, supervision, fringe benefits, contingent rewards (performance-based rewards), operating conditions (required rules and procedures), co-workers, nature of work and communication.

Four items assess each of these scales, and the total score was calculated from all items. A Likert scale with six choices was used, ranging from 1 (disagree) to 6 (agree). The items were written in both directions, so approximately half should be marked backwards. Scores on each of the nine scales ranged from 4 to 24. Negatively worded items were reversed before summing.

JSS scores were interpreted as follows: For the total of 36 items, scores were possible between 36 and 216, scores between 36–108 indicated dissatisfaction, ambivalence 109–143 and job satisfaction 144–216 [21].

The questionnaire was developed by Professor Paul Spector and the reliability values for the JSS were 0.91 and the range for the subscales was 0.60–0.82. It has also been applied to a population of Romanian origin and was translated at the Faculty of Psychology of Babeș-Bolyai University in Cluj-Napoca, Romania, by Professor Horia Pitariu [22].

This questionnaire was considered to be complete enough to be applied in our study.

The socio-demographic data collected were represented by age, gender, professional category, work experience, working hours and number of patients cared for per shift.

### 2.3. Data Analysis

The recorded data were framed in sampling lists, on which centralizing tables were then made. Final data analysis was performed in IBM SPSS Statistics version 20.0 and Microsoft Excel 2007. Descriptive statistics and correlations were performed with all the study variables. We used Pearson’s correlation to assess the relationship between burnout scales and job satisfaction, and significance was considered at the *p* < 0.05 level and a 95% confidence interval. To determine if the data were normally distributed, we used normality tests.

A Shapiro-Wilk’s test (*p* > 0.05) and a visual inspection of the histograms, normal Q-Q plots showed that job satisfaction and burnout scale scores were approximately normally distributed for nurses with a skewness of −0.320 (SE = 0.269) and kurtosis of −0.417 (SE = 0.532) for job satisfaction scores, for emotional exhaustion, the skewness was −0.047 ± 0.269 and the kurtosis was −0.105 ± 0.532, for depersonalization (skewness = −0.32 ± 0.269, Kurtosis = −0.417 ±0.532) and personal accomplishment (skewness = −0.164 ± 0.269, Kurtosis = −0.184 ±0.532)

In the framework of descriptive statistics, the values of location indicators (mean, median, ad modal value) and scattering (dispersion, standard deviation, and amplitude) were calculated.

## 3. Results

Of the total number of participants (*n* = 80), 59 (73.8%) were women. The ages of the participants ranged from 25 to 53 years, with an average of 37.7 years. Age was found to be bimodal. The majority of participants were 35–53 years old (68.7%) and 25–34 years old (31.3%). Average work experience was 8.8 years and 47.5% of nurses had between 5–10 years of professional experience in the field. Nurses worked 44.7 h per week on average and had 26.6 patients per shift. Level of education: Only 36.3% of nurses had a university degree. The gender distribution was 73.8% female and 26.3% male. (Table 1)

After two years, since the COVID-19 pandemic started in our country, only 36.25% (*n* = 29) of the nurses in the Emergency Department showed a high level of burnout. Regarding the burnout syndrome scales, the following results were obtained: Emotional exhaustion (EE) had an average value of 23.4 points (62.5%), depersonalization (DP) had an average value of 17.7 points (48.7%) and the low personal accomplishment scale (AP) had an average value of 27 points (60%). We can say that the burnout syndrome that manifested in the Emergency Department among nurses reached a high level (Table 2).

Next, we realized the bivariate correlation of the Pearson coefficient between the dimensions of the Burnout syndrome and the burnout criteria, and as we mentioned before, a person is considered to have burnout if he has a score of emotional exhaustion ≥27 plus a score of depersonalization ≥10. Statistically, all the coefficients were significant. Strong correlations were obtained between the emotional exhaustion dimension and the depersonalization dimensions (*r* = 0.841; *p* < 0.001) and low personal accomplishment (*r* = 0.735; *p* < 0.001). All these elements are significantly related to Burnout syndrome. Burnout related to emotional exhaustion is the scale that contributes most to total exhaustion (rp = −0.766; *p* < 0.001) (Table 3).

To highlight that the sex variable is an important indicator of Burnout syndrome, we conducted an ANOVA test. This uses, as a value, an independent scalar of the sex of the subjects (male/female). The descriptive results of the ANOVA test are presented in Table 4, and we found that the female subjects have higher average values of Burnout syndrome dimensions compared to male subjects (sig. 0.33). An important correlation of Burnout syndrome dimensions with professional experience in the Emergency Department was found. We showed that, in the first five years of experience, the EE dimension was on average 20.94 points, with a standard deviation of 6.46 points, and this variable raised to an average of 23.76 points with a standard deviation of 7.88 points in experienced subjects in the field for 5–10 years (sig. 0.24). For subjects with experience over 10 years, the EE dimension obtained an average of 24.86 points with a standard deviation of 8.34 points. Additionally, in the case of the DP (sig. 0.30) and AP (sig. 0.29) dimensions, an increase in the average value was observed in subjects with experience in the field ≥5 years. In the case of the age variable, higher scores are noted for all dimensions of the Burnout syndrome for the age range >36 years (sig. 0.22). In the case of the number of patients treated per work shift and the number of hours worked per week, no significant differences were revealed (Table 4).

Emergency Department nurses scored an average of 145.36 points, with 67.5% being satisfied with their work. Nurses were satisfied with most aspects of their work, including remuneration, supervision, rewards, operating conditions, colleagues and the nature of their work. Participants were ambivalent about promotion, fringe benefits and communication. The results of the study showed that Emergency Department nurses were satisfied with their work. Of the nine job satisfaction scales identified by Paul E. Spector, fringe benefits, promotion and communication scored a weighted mean of 13.57, 13.96 and 13.6, respectively, indicating that participants were ambivalent about the chance of promotion, communication with colleagues or superiors and job benefits. The nature of the work scale representing nurses’ tasks and responsibilities scored an average of 17.35, which means that nurses are satisfied with their type of work. (Table 5)

A Pearson correlation coefficient was calculated to assess the relationship between burnout scales and job satisfaction. There was a moderate relationship between emotional exhaustion and nature of work (*r* = 0.27; *n* = 80; *p* = 0.012) and job satisfaction (*r* = 0.25; *n* = 80; *p* = 0.026).

There was a strong relationship between depersonalization and nature of work (*r* = 0.29; *n* = 80; *p* = 0.008) and a moderate relationship with job satisfaction (*r* = 0.23; *n* = 80; *p* = 0.04) and rewards (*r* = 0.22; *n* = 80; *p* = 0.04). Increased perceptions of depersonalization were correlated with increased job satisfaction. Personal accomplishments were relatively proportional to the nature of work *r* = 0.26; *n* = 80; *p* = 0.018).

Increased job satisfaction was positively associated with the nature of work (*r* = 0.447; *p* < 0.001) and rewards (*r* = 0.419; *p* < 0.001) (Table 6).

An increased level of job satisfaction was positively correlated with job satisfaction and communication. Job promotion was negatively associated with supervision (*r* = 0.315; *n* = 80; *p* = 0.004), and moderately positively associated with job satisfaction (*r* = 0.028; *n* = 80; *p* = 0.012) (Table 7).

## 4. Discussion

The main objective of this research was to identify the prevalence of burnout syndrome among nurses in the Emergency Department, as well as the impact of burnout syndrome on job satisfaction.

After two years, since the start of the COVID-19 pandemic in our country, the results show a high risk of Burnout syndrome among the medical staff in the Emergency Department, with approximately one third of the subjects presenting a high risk of burnout and showing increased values in all Burnout syndrome dimensions. The data demonstrate a high prevalence of burnout in frontline nurses, with similar values reported in the literature [14,15,16,17,18,19,20,21,22,23,24]. The overall prevalence of Burnout syndrome among nurses is high, proving that the threat posed by the Covid-19 pandemic is a major stressor for nurses [25,26]. In this research, we found significant correlation coefficients between burnout dimensions and burnout criteria. The three dimensions are significantly related to the Burnout syndrome. Burnout related to emotional exhaustion is the dimension that contributes the most to the degree of exhaustion.

As a secondary objective, we thought to identify the correlation between the dimensions of the Burnout syndrome and the collected socio-demographic data.

Regarding experience in the field, we showed that burnout is correlated with the number of years worked in the Emergency Department. However, some studies have shown that medical personnel with more experience in the field can successfully manage periods of stress and feel safe at work [14,27].

In the case of the age variable, higher scores are noted for all dimensions of the Burnout syndrome for the age range >36 years, which corresponds to the data found in the literature [28].

In some studies, age is mentioned as a factor that does not interfere with the presence and levels of Burnout syndrome [29].

Analyzing the dimensions of the Burnout syndrome according to the gender variable, we found much higher average values in the case of female subjects [30].

The Emergency Department nurses show a high level of job satisfaction, being satisfied with remuneration, supervision, rewards, working conditions and their relationships with colleagues. Thus, our main hypothesis is confirmed.

Significant correlations were obtained between the dimensions of the Burnout syndrome and the scales of professional satisfaction, which confirms that there is a positive relationship between the two aspects and nurses who present a high risk of burnout associate increased professional satisfaction.

Other studies show that burnout has a negative impact on job satisfaction among healthcare professionals and may lead to negative outcomes of care rationalization [31,32].

Nurses in Emergency Departments are physically and mentally stressed due to the high volume of patients with various pathologies, poor working conditions (working in the pre-hospital, and high noise level), lifting weights (patient, stretcher and other medical equipment), and the nature of the work favoring the onset of burnout [33].

Of the nine job satisfaction scales identified by Paul E. Spector, fringe benefits, promotion and communication indicate that participants were ambivalent about the chance of promotion, communication with colleagues or superiors and job benefits. According to Albaugh, these influence job stresses [34].

Other studies claim that caring for patients brings the most job satisfaction but being ambivalent does not ensure that patients will receive quality medical care [35,36].

The nature of the work scale representing nurses’ tasks and responsibilities has a high score, which means that nurses are satisfied with their type of work.

The nature of work is one of the most important scales of job satisfaction. So long as nurses are satisfied with their type of work, patients will receive optimal care, with higher job productivity and lower turnover rates [37]. No correlations were found between job satisfaction and sociodemographic data. Emotional exhaustion, a component of burnout, is a predictor of increased job satisfaction among nurses working in the Emergency Department.

### Limitations and Recommendations for Practice and Futureresearch

This study has potential limitations. There are major limitations in this study that could be addressed in future research.

First, the sample chosen is a limited one and does not reflect the whole picture of the Burnout syndrome among the nurses in the Emergency Department. Further research on a larger sample is needed.

Second, the subject recruitment period was a limited one (March-April), with the work schedule of the assistants being a busy one, namely working on night shifts.

Third, in Romania, there are not many studies about the Burnout syndrome of nurses working in the Emergency Department, therefore this study has not only a scientific value but also a practical one. After two years of the COVID-19 pandemic, the medical staff from the Emergency Departments show a certain degree of burnout, and prompting intervention is necessary in their case.

To sum up, our results concerning the impact of Burnout syndrome on professional satisfaction can be informative for hospital managers to implement prevention and intervention strategies for professional burnout among front-line medical staff.

## 5. Conclusions

The results of this study suggest the relation between variables, and this could be used to implement psychoactive intervention strategies at both individual and organizational levels, which could lead to a decrease in burnout levels.

There is a high prevalence of burnout among nurses in the Emergency Department. The degree of impairment is significant and one third of nurses suffer from a high level of burnout. Nurses were satisfied with most aspects of their work, including remuneration, supervision, rewards, operating conditions, colleagues and nature of work. Participants were ambivalent about promotion, fringe benefits and communication. Emotional exhaustion, a component of burnout, is a predictor of increased job satisfaction among nurses working in the Emergency Department. Personal achievement was relatively commensurate with the nature of the work. Additionally, the increase in burnout among nurses is directly proportional to the nature of the work.

## Figures and Tables

**Table 1 medicina-58-01516-t001:** Demographic and professional characteristics of participants (*n* = 80).

Characteristics	*n*	%
Gender		
Male	21	26.3
Female	59	73.8
Age		
<35 years	25	31.3
>35 years	55	68.7
Years of Work Experiences		
<5 years	19	23.8
5–10 years	38	47.5
>10 years	23	28.7
Education		
Post -secondary studies	51	63.7
University studies	29	36.3
Number of worked hours/week	44.7	
Patients/shift	26.6	

**Table 2 medicina-58-01516-t002:** MBI-HSS scores; Nurses *N* = 80.

Emotional exhaustion (EE)	≤18	19–26	≥27
	13 (16.25%)	38 (47.5%)	29 (36.2%)
Depersonalization (DP)	≤5	6–9	≥10
	0	9 (11.25%)	71 (88.75%)
Personal accomplishment (PA)	≥40	39–34	≤33
	8 (10%)	6 (7.5%)	66 (82.5%)
Burnout criteria: EE ≥ 27 si/sau DP ≥ 10)	yes	no	
	29 (36.25%)	51 (63.75%)	

Emotional exhaustion (EE) ≥ 27, Depersonalization (DP) ≥ 10, Personal accomplishment (PA) ≤ 33, Burnout criteria EE ≥ 27 + DP ≥ 10.

**Table 3 medicina-58-01516-t003:** The correlation between burnout criteria, emotional exhaustion, depersonalization and personal accomplishment.

		EE	DP	PA	Burnout Criteria
EE	Pearson Correlation	1	0.841 **	0.735 **	−0.766 **
	Sig. (2-tailed)		0.000	0.000	0.000
DP	Pearson Correlation	0.841 **	1	0.806 **	−0.592 **
	Sig. (2-tailed)	0.000		0.000	0.000
PA	Pearson Correlation	0.735 **	0.806 **	1	−0.570 **
	Sig. (2-tailed)	0.000	0.000		0.000
Burnout criteria	Pearson Correlation	−0.766 **	−0.592 **	−0.570 *	1
	Sig. (2-tailed)	0.000	0.000	0.000	

* Correlation is significant at the 0.05 level and ** correlation is significant at the 0.01 level (2-tailed).

**Table 4 medicina-58-01516-t004:** Descriptive results of ANOVA testing for variable work experience, age and sex.

		*N*	Mean	95% Confidence Interval for Mean		Minimum	Maximum
				Lower Bound	Upper Bound		
	Work Experience						
EE	<5 years	19	20.94	17.83	24.06	9	29
	5–10 years	38	23.76	21.17	26.35	9	42
	>10 years	23	24.86	21.26	28.47	9	40
	Total	80	23.41	21.68	25.13	9	40
DP	<5 years	19	16.63	13.96	19.29	6	26
	5–10 years	38	17.34	15.24	19.44	6	33
	>10 years	23	19.43	16.54	22.32	6	28
	Total	80	17.77	16.37	19.17	6	33
PA	<5 years	19	25.21	22.32	28.09	10	33
	5–10 years	38	26.57	23.64	29.51	10	45
	>10 years	23	29.21	25.06	33.36	10	45
	Total	80	27.01	25.10	28.92	10	45
	Age						
EE	25–35	34	21.11	18.37	23.85	9	40
	>36	46	25.10	22.93	27.28	9	42
DP	25–35	34	15.91	13.79	18.03	6	28
	>36	46	19.15	17.32	20.98	6	33
PA	25–35	34	24.52	21.78	27.27	10	45
	>36	46	28.84	26.26	31.42	10	45
	Sex						
EE	male	21	22	18.09	25.9	9	35
	female	59	23.91	21.97	25.86	9	42
	Total	80	23.41	21.68	25.13	9	42
DP	male	21	15.90	12.46	19.34	6	28
	female	59	18.44	16.95	19.93	6	33
	Total	80	17.77	16.37	19.17	6	33
PA	male	21	24.9	21.24	28.56	10	40
	female	59	27.76	25.49	30.02	10	45
	Total	80	27.01	25.1	28.92	10	45

**Table 5 medicina-58-01516-t005:** Descriptive statistics of scales of JSS.

	N	Range	Minimum	Maximum	Mean	Std. Deviation	Skewness		Kurtosis	
	Statistic	Statistic	Statistic	Statistic	Statistic	Statistic	Statistic	Std. Error	Statistic	Std. Error
Pay	80	10	13.00	23	17.46	2.03	0.39	0.26	0.018	0.53
Promotion	80	8	10.00	18	13.96	1.918	−0.13	0.26	−0.481	0.53
Supervision	80	11	14	24	16.80	1.98	1	0.26	2.13	0.53
Fringe benefits	80	9	10	19	13.57	1.88	0.32	0.26	−0.26	0.53
Contingent rewards	80	9	12	21	16.57	1.86	−0.07	0.26	−0.048	0.53
Operating condition	80	7	12	19	16.78	1.42	−0.68	0.26	0.465	0.53
Coworkers	80	8	15	23	19.25	1.93	−0.36	0.26	−0.55	0.53
Nature of work	80	9	12	21	17.35	1.80	−0.75	0.26	0.59	0.53
communication	80	7	11	18	13.60	1.84	0.43	0.26	−0.66	0.53
Total satisfaction	80	26	130	156	145.36	4.80	−0.24	0.26	0.54	0.53
Valid N (listwise)	80									

**Table 6 medicina-58-01516-t006:** Correlations between burnout scales and job satisfaction, nature of work and fringe benefits.

		EE	DP	AP	Contingent Rewards	Nature of Work	Total Satisfaction
EE	Pearson Correlation	1	0.841 **	0.735 **	0.171	0.279 *	0.250 *
	Sig. (2-tailed)		0.000	0.000	0.129	0.012	0.026
DP	Pearson Correlation	0.841 **	1	0.806 **	0.225 *	0.295 **	0.230 *
	Sig. (2-tailed)	0.000		0.000	0.045	0.008	0.040
RR	Pearson Correlation	0.735 **	0.806 **	1	0.127	0.264 *	0.086
	Sig. (2-tailed)	0.000	0.000		0.262	0.018	0.447

* Correlation is significant at the 0.05 level and ** correlation is significant at the 0.01 level (2-tailed).

**Table 7 medicina-58-01516-t007:** Correlations between job satisfaction scales.

*N* = 80		Contingent Rewards	Nature of Work	Pay	Fringe Benefits	Promotion	Supervision	Communication	Total Satisfaction
Contingent rewards	Pearson Correlation	1	0.414 **	0.089	−0.142	−0.054	0.014	0.082	0.419 **
	Sig. (2-tailed)		0.000	0.433	0.208	0.635	0.899	0.468	0.000
Nature of work	Pearson Correlation		1	−0.027	0.041	−0.003	−0.101	−0.056	0.447 **
	Sig. (2-tailed)			0.809	0.720	0.976	0.374	0.620	0.000
pay	Pearson Correlation			1	−0.239 *	0.053	−0.024	0.087	0.361 **
	Sig. (2-tailed)				0.033	0.640	0.834	0.444	0.001
Fringe benefits	Pearson Correlation				1	0.178	−0.206	0.016	0.188
	Sig. (2-tailed)					0.114	0.066	0.888	0.095
promotion	Pearson Correlation					1	−0.315 **	0.071	0.280 *
	Sig. (2-tailed)						0.004	0.533	0.012
supervision	Pearson Correlation						1	−0.115	0.123
	Sig. (2-tailed)							0.308	0.276
communication	Pearson Correlation							1	0.399 **
	Sig. (2-tailed)								0.000
total_ satisfaction	Pearson Correlation								1
	Sig. (2-tailed)								

* Correlation is significant at the 0.05 level and ** correlation is significant at the 0.01 level (2-tailed).

## Data Availability

Informed consent was obtained from all subjects involved in the study.

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
