# Peer review of "The Impact of Burnout Syndrome on Job Satisfaction among Emergency Department Nurses of Emergency Clinical County Hospital “Sfântul Apostol Andrei” of Galati, Romania"

_medicina, 2022, doi:10.3390/medicina58111516_

Round 1

Reviewer 1 Report (Previous Reviewer 1)

I congratulate the authors on their revisions.

Minor issues persist.

Recommend adding "in Galata, Romania" to the title because the sample population is not representative of all nurses in Romania.

Once again, there is no such thing as low, medium, or high levels of burnout. Maslach has clearly stated this in her manual that attempts to differentiate scores into these categories are not evidence based. E.g. An EE score of 32 signifies a high probability of emotional exhaustion while an EE score of 5 signifies NO emotional exhaustion and not a low level of emotional exhaustion. Figures must be amended accordingly.

Author Response

Reviewer 1

Comments and Suggestions for Authors

I congratulate the authors on their revisions.

Minor issues persist.

Recommend adding "in Galata, Romania" to the title because the sample population is not representative of all nurses in Romania.

Once again, there is no such thing as low, medium, or high levels of burnout. Maslach has clearly stated this in her manual that attempts to differentiate scores into these categories are not evidence based. E.g. An EE score of 32 signifies a high probability of emotional exhaustion while an EE score of 5 signifies NO emotional exhaustion and not a low level of emotional exhaustion. Figures must be amended accordingly.

We have modified this in the text, according to Fourth Edition – MBI. The score of each dimension can be calculated using two methods. The first method sums up the scores by obtaining scores for each dimension and the second method calculates the average of the scores for each dimension. This study adopted the first method of calculation, so for each scale, the sum of the points is calculated, obtaining a score. Thus, emotional exhaustion is high for a score of at least 27, depersonalization is high for a score of at least 10, while personal accomplishment is low for a score of 33 or less.

Reviewer 2 Report (Previous Reviewer 2)

Dear authors

After reviewing your study, I will suggest some changes in order to improve the quality of your manuscript.

1.- Methodology: Please change "2.2 Tools" by "2.2. Instruments" (this is the correct term when we talk about psychology).

2.- Methodology. It is not necessary include information about who develop a questionnaire or who translated. When you define the instruments used in your study, please include the information about the last questionnaire validated to the population. Including the information about the questionnaire reliability. 

3.- Methodology. When you write:  "This questionnaire was developed for all types of organizations. Given the nine scales it develops, it is considered to be complete enough to be applied to the present study", what is your point here? 

Please, eliminated this sentence the justification of your selection it couldn't be supported by this.

4.- Methodology-Data Analysis. Please include information about the application of normality tests in your sample.

Furthermore, please, include a clarification about the reason why you use a Pearson's correlation when your numeric data come from questionnaires.

5. Results

Why do you include in table 4, the standard deviation and standard error?

What extra information is offered for the inclusion of these 2 data?.

 6.- Discussion

Please, include the limitation of your study at the end of your discussion.

7.- Your study is an observational study, you can't conclude that your results "demonstrate",  your study "suggests the relation between variables and this could be used to ..."

Author Response

Reviwer 2

Comments and Suggestions for Authors

Dear authors

After reviewing your study, I will suggest some changes in order to improve the quality of your manuscript.

1.- Methodology: Please change "2.2 Tools" by "2.2. Instruments" (this is the correct term when we talk about psychology).

 We did that.

2.- Methodology. It is not necessary include information about who develop a questionnaire or who translated. When you define the instruments used in your study, please include the information about the last questionnaire validated to the population. Including the information about the questionnaire reliability. 

This instrument has also been applied to a population of Romanian origin. For this study we have determined the Cronbach’s alpha coefficient with values 0.79 for MBI-HSS.

Other reviewer wanted to include the information about who develop the questionnaire or who translated.

3.- Methodology. When you write:  "This questionnaire was developed for all types of organizations. Given the nine scales it develops, it is considered to be complete enough to be applied to the present study", what is your point here? 

Please, eliminated this sentence the justification of your selection it couldn't be supported by this.

 We eliminated that

4.- Methodology-Data Analysis. Please include information about the application of normality tests in your sample.

Furthermore, please, include a clarification about the reason why you use a Pearson's correlation when your numeric data come from questionnaires.

We use Pearson’s correlation to assess the relationship between burnout scales and job satisfaction. To determine if the data were normally distributed, we used normality tests.

A Shapiro-Wilk’s test (p>.05) and a visual inspection of thei histograms, normal Q-Q plots showed that job satisfaction and burnout scale scores were approximately normaly distributed for nurses with a skewness of  -0.320 (SE= 0.269) and kurtosis of -0.417 (SE= 0.532) for job satisfaction scores,  for emotional exhaustion (skewness=  -0.047± 0.269 and kurtosis=-0.105±0.532, for depersonalization (skewness = -0.32 ± 0.269, Kurtosis = -0.417 ±0.532) and personal accomplishment (skewness = -0.164 ± 0.269, Kurtosis = -0.184 ±0.532)

  1. Results

Why do you include in table 4, the standard deviation and standard error?

What extra information is offered for the inclusion of these 2 data?.

 We removed the standard deviation and standard error to simplify the table.

 6.- Discussion

Please, include the limitation of your study at the end of your discussion.

 We did that.

7.- Your study is an observational study, you can't conclude that your results "demonstrate",  your study "suggests the relation between variables and this could be used to ..."

 We dit that.

This manuscript is a resubmission of an earlier submission. The following is a list of the peer review reports and author responses from that submission.

Round 1

Reviewer 1 Report

Thank you for giving me the opportunity to review the manuscript.

Moscu et al attempted to study burnout and its relation to job satisfaction in a population of nurses within an emergency department in a Romanian hospital.

Overall, the language is good but is in need of minor corrections. However, burnout in nurses have been studied before, so this study does not warrant publication because of the lack of novelty. It may be novel if burnout in Romanian ED nurses have not been reported before...

There are also typos in the illustrations e.g. "total scor".

Importantly, the most significant flaw is the methodology of the study. The authors used the MBI tool to measure burnout but do not adopt the criteria recommended by Maslach in the most recent MBI manual. The study for which the authors based their MBI cut-offs on is seriously flawed.

Also, it is impossible to determine burnout as "low", "medium", or "high" by using MBI scores. This has been acknowledged openly by C Maslach.

As a result, downstream analyses and the correlation studies are not interpretable.

A discussion section is also missing.

I wish the authors all the best in addressing these points so that the manuscript can be improved to a standard worthy of a publication.

Reviewer 2 Report

Dear authors

After reviewing your manuscript I have some commentaries to do in order to improve its quality. 

1.- Introduction. Is necessary that explain deeply the aim of your study. After reading your study's background, I found it incomplete. 

As an example, you don't explain correctly the dimensions that explore in your study. 

Please, review a re-write your introduction to explain deeply the actual knowledge about the dimensions analyzed in your study and the possible relation with nurses in emergency units.

2.- Aim of your study. Please, explain the Hypothesis that you pursuing to explore with your study. The explanation is so unspecific.

3.- Material and methods:

3.1. PLease, include information about the country where the study was carried out.

3.2. Is necessary to include information about the Research Ethic Committee, you have mentioned that is the "local" but, this committee must be associated with a University, Hospital, or Research Center. Please include this information.

3.3. Please, could you explain if the questionnaires used in your study were validated by your population? Have the questionnaires used been adjusted, and validated, to Romanian nurses?

This is the main problem that I find in your manuscript. I need an explanation about the questionnaires.

3.4. Data analysis

Could you explain how have you used Pearson's coefficient to measure the correlation between qualitative variables? Have you converted the qualitative variables into quantitative ones?

Why you don't use chi-square or Spearman's correlation coefficients? Please explain how you afford this in your "data analysis" section.

4. Results.

4.1. In your study's aim, explain that you'll explore the relationship between sociodemographic variables and burnout and job satisfaction.

But, it was impossible for to me find these analyses in the results section. PLease include the analysis.

 4.2. In the result section NEVER include references. Please, eliminate them (page 4, lines 144-145)

4.3. Could you explain the information that offers in figure 3 about the "distribution of job satisfaction"?

If you include, in the data analysis section, that you will do analysis to check the homoscedasticity and normality will be enough (indicating, of course, what kind of analysis you'll do)

And, results, like figure 3, will not be necessary.

4.4. Table 3 and Table 4, are confusing and could induce confusion in future readers. Please, review them and try to show the information included in them in an easier way.

Table 3. You include the foot inside the table. 

5. Discussion. 

5.1. Please, follow the official template of the MEdicine journal. The discussion must appear in a different section.

5.2. There are no "limitations" in your discussion. Please explain deeply the limitations of your study. (Sampling method, questionnaires, sample, your study design, etc.)